# *Libidibia ferrea* (jucá), a Traditional Anti-Inflammatory: A Study of Acute Toxicity in Adult and Embryos Zebrafish (*Danio rerio*)

**DOI:** 10.3390/ph12040175

**Published:** 2019-11-30

**Authors:** Diego Q. Ferreira, Thamara O. Ferraz, Raquel S. Araújo, Rodrigo Alves Souza Cruz, Caio Pinho Fernandes, Gisele C. Souza, Brenda L. S. Ortiz, Rosangela S. F. R. Sarquis, Jemima C. M. M. Miranda, Rafael Garrett, José C. Tavares Carvalho, Anna Eliza M. de Faria Mota Oliveira

**Affiliations:** 1Department of Biological and Health Sciences, Federal University of Amapá, Rodovia Juscelino Kubitisheck, km 02, Macapá, Amapá CEP 68902 280, Brazil; diegomendesmauer@hotmail.com (D.Q.F.); thamara_96@hotmail.com (T.O.F.); raquel.araujo@unifap.br (R.S.A.); rodrigo@unifap.br (R.A.S.C.); caiofernandes@unifap.br (C.P.F.); 2Laboratory of Phytopharmaceutical Nanobiotechnology, Department of Biological and Health Sciences, Federal University of Amapá, Rodovia Juscelino Kubitisheck, km 02, Macapá, Amapá CEP 68902 280, Brazil; 3Pharmaceutical Research Laboratory, Department of Biological and Health Sciences, Federal University of Amapá, Rodovia Juscelino Kubitisheck, km 02, Macapá, Amapá CEP 68902 280, Brazil; custodio_gisele@yahoo.com.br (G.C.S.); charmed1797@gmail.com (B.L.S.O.); rosangela.sarquis.bot@gmail.com (R.S.F.R.S.); 4Molecular Biology Laboratory, Department of Biological and Health Sciences, Federal University of Amapá, Rodovia Juscelino Kubitisheck, km 02, Macapá, Amapá CEP 68902 280, Brazil; jemima@unifap.br; 5Metabolomics Laboratory, Institute of Chemistry, Federal University of Rio de Janeiro, Rio de Janeiro RJ 21941-902, Brazil; rafael_garrett@iq.ufrj.br

**Keywords:** *Libidibia ferrea*, toxicity, zebrafish, histopathology, embryos

## Abstract

The plant species *Libidibia ferrea* (Mart. ex Tul.) LP Queiroz var. *ferrea* basionym of *Caesalpinia ferrea* (Mart. ex Tul.) is used in various regions of Brazil in folk medicine in the treatment of several health problems, especially in acute and chronic inflammatory processes. Most of the preparations employed are alcoholic. Therefore, this study aimed to evaluate the acute toxicity of the hydroethanolic extract of fruits of *Libidibia ferrea* (EHEFLf) in zebrafish, emphasizing the possible changes in the organic-cellular level of the gills, liver, kidneys, and intestine and on embryos. The result obtained by LC-M/MS from EHEFLf indicated a high concentration of possible polyhydroxylated substances. EHEFLf, at a dose of 2 g/kg orally, produced non-significant alterations of the analyzed organs. However, for embryos, the treatment with different concentrations demonstrated heart toxicity that was concentration-dependent. There is no evidence of a correlation of the observed effects with the phytochemical composition, and considering the species of animal used, it can be suggested that the oral use of *L. ferrea* hydroethanolic extract has an acceptable degree of safety for use as an oral medicinal product. and embryo results have shown significant affinity to the heart; however, it is perceived to be related to the concentrations used.

## 1. Introduction

Popular knowledge of traditional medicinal plants (TMPs) in the Amazon has resisted technological advancements within the pharmaceutical field mainly due to the presence of the indigenous population. However, the lack of scientific knowledge and/or the misuse of some TMP can lead to short- and long-term toxic effects on the human organism [1].

In the Brazilian Amazon, *Libidibia ferrea* is commonly used as TMPs to treat inflammations, infections, and hyperglycemia. *L. ferrea* is a native and endemic tree species from Brazil, belonging to the Fabaceae family. It is present in several phytogeographic domains such as Caatinga, Cerrado, and the Atlantic Forest. Its distribution is registered throughout the Brazilian territory, mainly in the Brazilian Amazon and Northeast [2,3,4]. Popularly known as jucá or pau-ferro, *L. ferrea* is widely used in traditional medicine due to the diversity of bioactive compounds such as terpenoids and phenolic compounds present in different parts of the vegetable such as bark, fruits, seeds, leaves, and roots [5].

Several biological activities from the fruits and bark of *L. ferrea* have been reported in the literature as anti-inflammatory [6], gastric anti-ulcer [7], antidiabetic [8], healing [9], antitumor [10], anti-proliferative [11], hepatoprotective [12], antimicrobial [13], and in vitro leishmanicidal [14]. However, despite the widespread use of popular alcoholic preparations and pharmaceutical dispensing of products with derivatives of this plant species, few studies have reported the safety of these products [15,16,17], especially in an alternative model such as zebrafish. Therefore, this study evaluated the action of oral EHEFLf on adults and embryos *Danio rerio* (zebrafish), emphasizing histopathological aspects and morphofunctional changes.

## 2. Results

The negative-ESI Liquid Chromatography-High Resolution Tandem Mass Spectrometry (LC-HRMS/MS) analysis of EHEFLf revealed the presence of phenolic and sugar compounds (Table 1). Their identification was performed by comparing their high accurate *m*/*z* ions with a mass error below 5 ppm and MS/MS fragmentation pattern to those available in the literature (Mass Bank of North America-MoNA; https://mona.fiehnlab.ucdavis.edu/). Compounds of retention time (Rt) 2.1 and 4.0 were not identified, but it would be possible to assign a molecular formula of C_21_H_10_O_13_ and C_9_H_10_O_5_, respectively (Figure 1).

The compound of Rt 4.7 presented high abundance in the chromatogram and was tentatively identified as ellagic acid. Previous work in the literature has described the presence of lipids, phenolic substances, and sugars in *L. ferrea* [18]. The compound of Rt 4.7 presented high abundance in the chromatogram and was tentatively identified as ellagic acid.

In the assessment of acute toxicity in adult zebrafish, no behavior-related changes were observed at 48 h of observation in either the EHEFLf-treated or the 1% propylene glycol-treated control group. Regarding the histopathological analysis performed in different tissues, it is possible to describe, in general, that in the gills and intestine, the treatment produced slight alterations. However, in the kidneys and liver, significant alterations were found when compared to the propylene glycol 1% (control group), which indicates that the phytochemical markers present in EHEFLf were highly metabolized, influencing the cellular structure of these organs (Figure 2).

At the gill level, it was observed that all findings fit the level I alteration, obtaining an average HAI of 6.4 ± 0.5 and 3.2 ± 0.3 for EHEFLf and the control group, respectively, indicating mild alteration (Table 2 and Figure 3).

Concerning the liver, the treatment with EHEFLf produced levels I and II HAI, showing no significant difference when compared to the control group that presented level I HAI (Table 3 and Figure 4).

Regarding the EHEFLf group, the histopathological findings of the intestine observed an HAI that fit levels I and II, classified as mild to moderate alterations, with a mean of 15.2 ± 2.0; a similar fact was observed for the control group with levels I and II HAI, and a mean of 12.2 ± 0.5 (Table 4 and Figure 5).

Oral EHEFLf treatment produced histopathological alterations with HAI with an average of 28.6 ± 2.6 that fit the level I and II alterations, classified as moderate to severe alterations, with an emphasis on changes in the glomerulus and tubular level. Already, the control group treated with propylene glycol 1%, presented histopathological alterations that fit the alterations of levels I and II, classified as normal, with a HAI of 6.4 ± 2.3 (Table 5 and Figure 6), without alterations to the glomerular level and Bowman capsule.

Regarding the effects of EHEFLf on embryos, there have been no previous reports, so the teratogenic alterations and lethal effect (coagulation and absence of cardiofetal beats) presented by the embryos upon contact with solutions containing EHEFLf were evaluated. It was observed that 1% propylene glycol solution and EHEFLf concentrations of 25, 50, and 125 mg/L had a lethal effect on embryos of 26.6%, 30%, 33.3%, and 10%, respectively (Table 6). All solutions showed no morphological/teratogenic changes on survival embryos, except for EHEFLf at 250 and 500 mg/L, which demonstrated cardiac edema, yolk sac edema, and scoliosis (Table 6 and Figure 7).

## 3. Discussion

The use in folk medicine of the species *Libidibia ferrea* in Brazil is of great significance, being that the bark, fruits, and leaves of this species are used in alcoholic and non-alcoholic preparations. In this study, we explored the fruit extract, which is widely used in folk medicine for asthma and various inflammatory states [6,19]. This activity can be due to the presence of bioactive compounds in the plant. The phytochemical composition of *L. ferrea* is poorly explored when compared to the biological effects of this plant [6,20]. Carvalho et al. [6] demonstrated the anti-inflammatory activity of the aqueous extract of *L. ferrea* fruits administered orally at a dose of 300 mg/kg in Wistar rats, by inhibiting the induced hind paw edema. Lima et al. [20] were able to demonstrate the inflammatory inhibition reactions and pain in Swiss albino mice by using the ethanolic extract of the fruits of *L. ferrea*, probably due to a reduction in vascular permeability. Pereira et al. [21] isolated three major polysaccharide fractions (FI, FII, FIII) of *L. ferrea* by chromatography of TPL (with a 2.8% yield), to which they attributed the anti-inflammatory effect.

The chemical composition results of *L. ferrea* obtained in this work are in agreement with previous studies, since Sampaio et al. [13] detected polyphenols in fruit extracts at 7.3% and H1 NMR analysis revealed hydroxy phenols and methoxylated compounds. Nakamura et al. [10] identified gallic acid and methyl gallate as active constituents from *L. ferrea* fruits. Ueda et al. [8] identified ellagic acid as a major compound from the fruit extract, which was elucidated as 2-(2,3,6-trihydroxy-4-carboxyphenyl) ellagic acid. Gallic and ellagic acid are polyphenols and are considered to be the main phytochemical components of *L. ferrea* fruits acting in the signaling pathway of pain, inflammation, and oxidative stress [22]. Araújo et al. [23] revealed in HPLC analysis the considerable presence of phenolic compounds: condensed tannins (catechins) and hydrolysable tannins (gallic acid), as well as demonstrating strong anti-inflammatory activity. Ferreira et al. [24] investigated the phenolic variability in the fruit of *L. ferrea* by ultraviolet/visible and chromatographic methods. They demonstrated that the polyphenolic content ranged from 13.99 to 37.86 g%, expressed as ellagic acid, or from 10.75 to 29.09 g%, expressed as ellagic acid by UV/VIS. The contents of ellagic and gallic acid determined by the liquid chromatography-reversed phase method ranged from 0.57 to 2.68 g% and from 0.54 to 3.23 g%, respectively.

Regarding toxicity, the toxicity of *L. ferrea* fruits [15,16,17,24] in mice or rat experimental models following OECD recommendations using a single dose, has been reported in the literature. However, some studies did not show the animal mortality, their behavior, or the histopathology as we have performed here for the first time with the alternative model, *D. rerio* (zebrafish). Therefore, this study is of great importance to study the safety of this species, as studies in zebrafish are currently considered by government agencies for the registration of new drugs.

It is important to consider the histopathological changes produced by drugs at the level of the fish gills in an attempt to expel the toxic/stressful substance, as they may hyperventilate, increasing the opercular movement, and consequently overloading the gill tissue, causing damage. Therefore, these alterations may indicate an injury to other organs directly related to drug metabolism such as the liver and kidneys [25,26]. The oral dose of 2 g/kg of EHEFLf produced no significant histopathological changes in the gills.

The liver is the organ responsible for some essential functions in the organism, and alterations in this organ may indicate damage related to metabolism, biotransformation, and excretion of certain substances [27,28]. Once liver damage is established, the homeostasis of the organism as a whole can be affected by interfering with some of the organ’s basic functions such as the synthesis and biotransformation of molecules [29]. Therefore, liver histology is an important biomarker for evaluating the effects of xenobiotics [30]. One of the histopathological findings when analyzing the group treated with EHELf was cytoplasmic vacuolization, a fact that was also shown in the study by Gayão et al. [31], who attributed this phenomenon to the possible accumulation of lipids, thus impairing liver metabolism.

The kidneys are of great importance for the metabolism of drugs in the human body, since they play a fundamental role in their elimination as well as their metabolites, and can directly influence the half-life of these substances in the body. In animal species, the kidneys play a major role in the control of homeostasis, and urine is the main biological fluid responsible for the excretion of the vast majority of drugs. This process of elimination can take place in three ways: glomerular filtration, active tubular secretion, and passive tubular reabsorption [32,33]. Luminal structural alteration in the kidneys such as tubular lumen enlargement, may be related to hyperchromasia, eosinophilia, and cellular apoptosis, which may lead to loss of renal function and consequently disordered proliferation of tubular cells as a compensatory mechanism to repair loss, possibly caused by a toxic factor [34,35]. In this context, lesions related to the renal lumen structure may be directly related to the progression to necrosis of these cells [36]. Changes related to the increase and decrease of Bowman’s capsule may impair glomerular filtration levels [36,37]. In addition to filtration, this type of alteration can also impair excretion rates, thus leading to excessive glomerulus production, which is not normal in this species [36]. Studies conducted by Souza [17] using a dose of 2 g/kg orally with an aqueous extract of *L. ferrea* fruits revealed liver and kidney changes similar to those found in this study.

Pickler et al. [38] studied a hydroalcoholic extract of bark and seed of *L. ferrea* that had a phytochemical marker of ellagic acid, and at a dose of 1 g/kg orally, concluded that extracts of *L. ferrea* do not exhibit safety levels compatible with being used in the gestational period.

A study by Falcão et al. [39], with a hydroethanolic extract from L. ferrea fruits, detected the presence of gallic acid and ellagic acid and confirmed the anti-inflammatory, antioxidant, and antinociceptive activities in vivo and enhanced cell viability in vitro.

It is important to highlight the possible relationship between the tissue-cellular alterations found in this study and the phytochemical composition of the EHEFLf. Despite the fact that phenolic compounds have great prominence producing antioxidant and anti-inflammatory effect, these compounds depend on complex metabolism requiring extensive hepatic and renal activity, which may result in visible, reversible tissue damage depending on the time of use [34]. However, it should be considered that in this study, the maximum dose of 2 g/kg, which did not produce mortality in zebrafish, and was previously used by Souza [17] in a study of dyslipidemia and subchronic toxicity.

Souza et al. [40] also used the absence of beating and egg coagulation as a parameter to account for the lethal effect on embryos, and evidenced cardiac edema, yolk sac edema, yolk edema, tail deformation, and scoliosis as teratogenic alterations in their study. Bittencourt et al. [41] related future cardiovascular problems with teratogenic changes in the heart of embryos, which was similar in this study when increasing the dose of EHEFLf. Sun et al. [42], when studying the toxic effects of butyl benzyl phthalate on embryos, also showed the same teratogenic alterations, but mainly emphasized the cardiac ones, showing that the toxic effects can genetically influence cardiac alterations in this model.

## 4. Materials and Methods 

### 4.1. Botanical Material

The botanical material, consisting of the aerial part with fruits of *Libidibia ferrea*, was collected in the municipality of Macapá, Amapá, Brazil, in the Medicinal Plants garden of the Institute of Scientific and Technological Research of Amapá-IEPA, under the geographic coordinates N00°02′49.4″ and W51.03′40.5″.

The identification of plant material was performed by a comparison with exsiccates deposited in a regional herbarium by botanist Dra. Rosângela do Socorro F. R. Sarquis. Samples of the species were deposited in the IAN Herbarium of Embrapa Eastern Amazon under the registration number 195949.

### 4.2. Obtaining EHEFLf

The collected *L. ferrea* fruits were placed in trays and submitted to oven drying, with the controlled temperature at 40 ± 2°C, for four days. After drying, the resulting material was manually cracked and ground in a knife mill and subjected to extraction by the 96% (v/v) ethanol percolation technique, with solvent renewal after 24, 12, and 6 h. In the first cycle, 1.25 L of solvent (96% ethanol) was added and 0.5 L in the second. After this period, the extract was concentrated in a rotary evaporator (Fisatom, Brazil) at 45°C and then placed under ventilation (48 h) to remove excess solvent. At the end of the process, the extract was lyophilized (Marconi Freeze Dryer, Brazil) and stored tightly closed under refrigeration.

### 4.3. EHEFLf Analysis by Liquid Chromatography-High Resolution Tandem Mass Spectrometry (LC-HRMS/MS)

EHEFLf was analyzed by LC-HRMS/MS using a Dionex UltiMate 3000 UHPLC system (Thermo Scientific, Bremen, Germany) coupled to a high-resolution QExactive Plus Orbitrap mass spectrometer (Thermo Fisher Scientific, Bremen, Germany) equipped with an electrospray ionization source operating in negative ion mode. Source ionization parameters were a spray voltage of 3.6 kV; capillary temperature of 300°C; S-Lens RF level 50, sheath and auxiliary gases 45 and 15 (arbitrary units), respectively. Samples were analyzed in the scan range of *m*/*z* 150 to 1000 at a resolution of 35,000 followed by data-dependent MS2 (ddMS2 experiments) using a resolution of 17,500 and normalized collision energy (NCE) stepped 15−35%. The column used was an ACE 3 C18 (150 mm × 2.1 mm × 3.0 µm) maintained at 40°C. The mobile phase consisted of (A) 0.1 % formic acid and (B) acetonitrile in gradient elution mode (0–1 min, 10% B; 1–16 min, 95% B; 16–19 min, 95% B; 19.1–24 min, 10% B) at a flow rate of 350 µL min^−1^. Compound identification was performed comparing the *m*/*z* ions with a mass error below 5 ppm and their MS/MS spectra to those available at the MassBank of North America (http://mona.fiehnlab.ucdavis.edu/) and in the literature.

### 4.4. Acute Toxicity Study in Zebrafish

#### 4.4.1. Animals

The animals used (*Danio rerio*) were from Aqua New Aquarius e Pisces Ltda. (PE, Brazil), male and female, of wild AB strain. They were stored in aquariums at the Zebrafish Platform of the Pharmaceuticals Research Laboratory of the Federal University of Amapá–UNIFAP, underwent an adaptation period (40 days), and were kept in temperature-controlled shelves (23 ± 2 °C) following a 12 h light/dark cycle (light period from 7:00 a.m. to 7:00 p.m.), and received flake commercial feed (Alcon Colors, Santa Catarina, Brazil) twice a day.

This study was approved by the Animal Experimentation Ethics Committee of the Federal University of Amapá (Brazil) under registration number 005/2018.

#### 4.4.2. Adult Zebrafish Study

The assessment of acute toxicity in adult zebrafish was established from an adaptation based on the guidelines described by the Organization for Economic Cooperation and Development (OECD) 425 OECD [43]. The toxicity test consisted of two steps: 2 g/kg oral limit test following the technique described by Borges et al. [26].

The initial test consisted of the administration of 2 g/kg of EHEFLf orally to a single animal. If the death occurred within 48 h, the main test was started. If it remained alive, the limit test was started. In the limit test, four animals were treated with 2 g/kg EHEFLf, so that the total number of animals tested was five (counting the test dose animal). Within 48 h, deaths were observed. In the presence of three or more deaths, the main test was started, otherwise, with three or more live animals, the tested EHEFLf is considered highly safe and the LD50 cannot be calculated.

#### 4.4.3. Behavioral Assessment

Behavior assessment followed the methodology described by Souza et al. [36], and the most intense human observation occurred in the first four hours, with observations every 30 min in the first four hours and thereafter every 12 h. The following parameters were recorded (Table 7).

#### 4.4.4. Study on Zebrafish Embryos

Embryotoxicity testing followed the recommendations of OECD 236 [44]. Two male and one female animals were kept in a breeding aquarium, separated by a partition, which was removed when the ambient lights were turned on so that eggs could be fertilized.

To avoid variants of the genetic type, it was decided to collect eggs from at least three groups, after washing with water from the aquarium where the parent fish were acclimatized, and the eggs were randomized and randomly selected, following the methodology described by Yang et al. [45]. Then, 30 eggs were transferred to each petri dish with dilutions of the EHEFLf-containing solutions and then one egg at a time was removed and placed in a pre-prepared 96-well plate containing 250 µl in each well of the following EHEFLf concentrations: 25, 50, 75, 125, 250, and 500 mg/L, in addition, to controls in water and 1% propylene glycol solution (vehicle) and subsequently incubated in the greenhouse (SOLAB SL-102/630, Brazil) at 28 ± 2°C up to 96 h after fertilization (hpf).

#### 4.4.5. Histopathological Study

The histopathological study was based on the techniques described by Souza et al. [46]. Gills, kidneys, liver, and intestines were evaluated. Euthanasia of the animals was performed following the methodology described by Favoretto et al. [47]. The material was stained according to the protocol of Souza et al. [36], and was stained with Harris Hematoxylin (Laborclin, Sinop, Mato Grosso, Brazil) and yellowish eosin (Inlab, Brazil). The histological slides were analyzed by an Olympus optical microscope (BX41-micronal, Brazil) and photographed with an MDCE-5C USB 2.0 (digital) camera.

The histopathological alterations index (HAI) was established from the alterations found in the gills, liver, kidneys, and intestine. After the calculation, it was classified as stage I, II, and III according to the alterations presented, being classified as: normal (0 to 10), moderately altered (11 to 20), moderate to severe (21 to 50), or severe irreversible (>100) [34,48,49,50]. The indices were calculated according to the following formula:(1)$$I=\frac{\sum_{i=1}^{na}a_{i}+10\sum_{i=1}^{nb}b_{i}+10^{2}\sum_{i=1}^{nc}c_{i}}{N}$$
where *a* = first stage;*b* = second stage;*c* = third stage;*na* = total number of changes considered to be first stage;*nb* = total number of changes considered to be second stage;*nc* = total number of changes considered to be third stage; and*N* = number of fish analyzed per treatment.

#### 4.4.6. Statistical Analysis

The results were expressed as mean ± standard deviation of the mean and for the statistical analysis, we used analysis of variance, ANOVA, followed by the Tukey test. Results with *p* < 0.05 were considered statistically significant.

## 5. Conclusions

From the results obtained in this study, it can be suggested that EHEFLf has an acceptable degree of safety for use as an oral medicinal product since an overestimated dose was used and changes in tissue-cellular level were considered, which was dose-dependent, even based on comparative terms with other studies. The results on embryos showed a significant heart toxicity, however, it was noticed that this was dependent on the EHEFLf concentration employed. Therefore, considering the species of animals already used in the experiment, it can be suggested that the oral use of the *L. ferrea* fruit product is safe.

## Figures and Tables

**Figure 1 pharmaceuticals-12-00175-f001:**
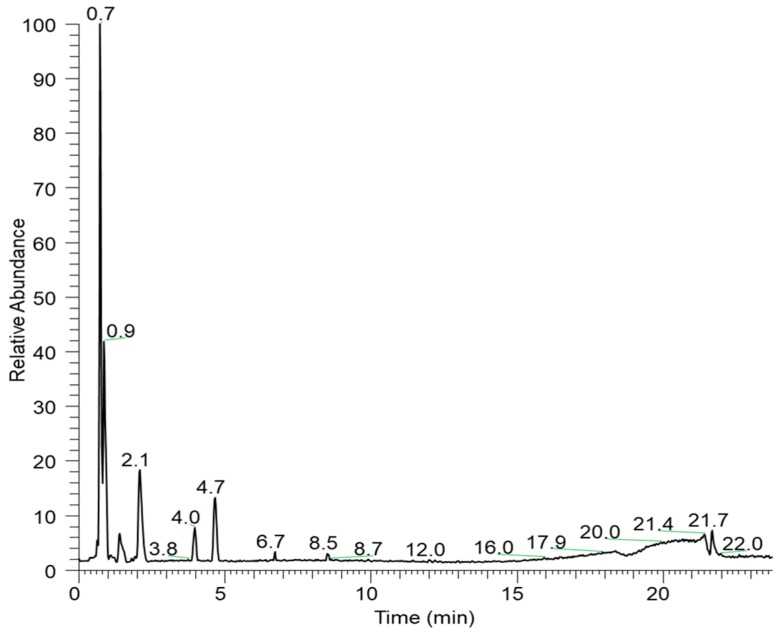
Base peak chromatogram of the hydroethanolic extract of fruits of *Libidibia ferrea* (EHEFLf) by Liquid Chromatography-High Resolution Tandem Mass Spectrometry (LC-HRMS/MS).

**Figure 2 pharmaceuticals-12-00175-f002:**
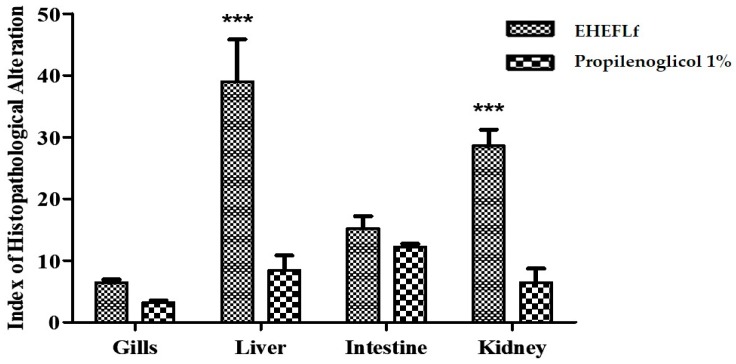
Histopathological alteration index (HAI) of *D. rerio* gills, liver, intestine, and kidneys, after 48 h of EHEFLf or propylene glycol oral administration at a dose of 2 g/kg.

**Figure 3 pharmaceuticals-12-00175-f003:**
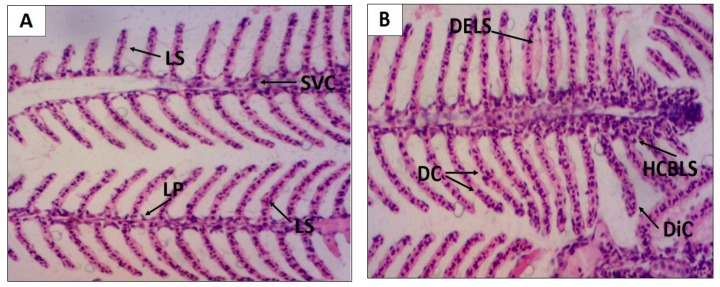
Histological alterations of *D. rerio* gills after oral treatment with 2 g/kg EHEFLf (**A**) and propylene glycol 1% (**B**). Gill filament of an animal treated with EHEFLf, where secondary lamellae (LS), primary lamellae (LP), and central venous sinus (SVC) were observed (**A**); gill filament of a vehicle-treated animal, where dislocation or elevation of secondary lamellae (DELS), capillary derangement (DC), capillary dilation (DiC), and epithelial cell hyperplasia at the base of secondary lamellae (HCBLS) were observed (**B**).

**Figure 4 pharmaceuticals-12-00175-f004:**
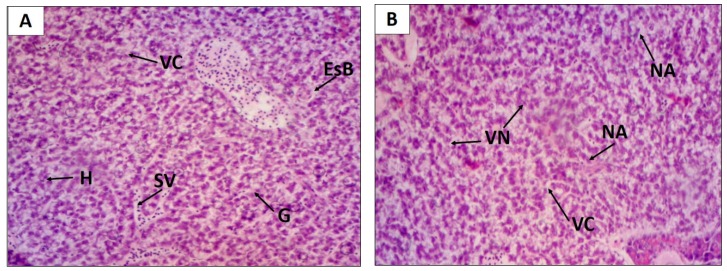
Histological changes of the *D. rerio* liver after oral treatment with 2 g/kg of EHEFLf (**A**) and propylene glycol 1% (**B**). Liver of an animal treated with EHEFLf, where hepatocytes (H), venous sinusoids (SV), glycogen (G), cytoplasmic vacuolization (VC), and biliary stagnation (EsB) were observed (**A**); liver of animal treated with vehicle where nuclear vacuolization (VN), cytoplasmic vacuolization (VC), and nuclear atrophy (NA) were observed (**B**).

**Figure 5 pharmaceuticals-12-00175-f005:**
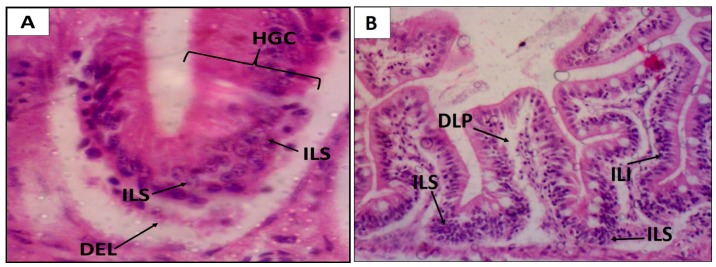
Histological changes of the *D. rerio* intestine after oral treatment with 2 g/kg of EHEFLf (**A**) and propylene glycol 1% (**B**). Intestine of an EHEFLf-treated animal where stromal lymphocytic infiltration (SLI), distal of the epithelial lining of the intestinal apex (DEL), leukocyte infiltration (ILI), and goblet cell hyperplasia (HGC) were observed (**A**); intestine of animal treated with vehicle-control, where stromal lymphocytic infiltration (ILS), lamina propria detachment (DLP), and leukocyte infiltration (ILI) were observed (**B**).

**Figure 6 pharmaceuticals-12-00175-f006:**
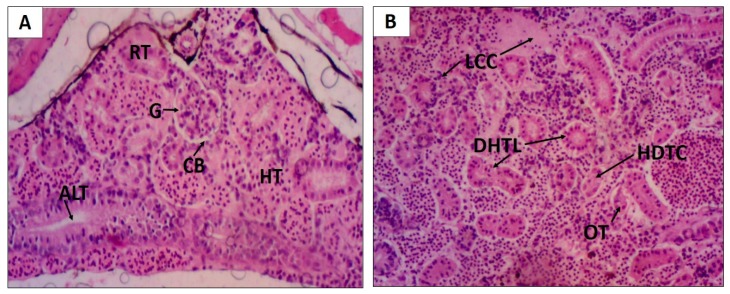
Histological changes of the *D. rerio* kidney after oral treatment with 2 g/kg of EHEFLf (**A**) and propylene glycol 1% (**B**). (**A**) Kidney of an EHEFLf-treated animal where renal tubules (RT), hematopoietic tissue (HT), glomerulus (G), Bowman capsule (CB), and tubular lumen enlargement (ALT) were observed; (**B**) kidney of animal treated with the vehicle-control where loss of the cellular contour (LCC), cytoplasmic tubular cell degeneration (CDTC), mild tubular hyaline degeneration (DHTL), and tubular obstruction (OT) were observed.

**Figure 7 pharmaceuticals-12-00175-f007:**
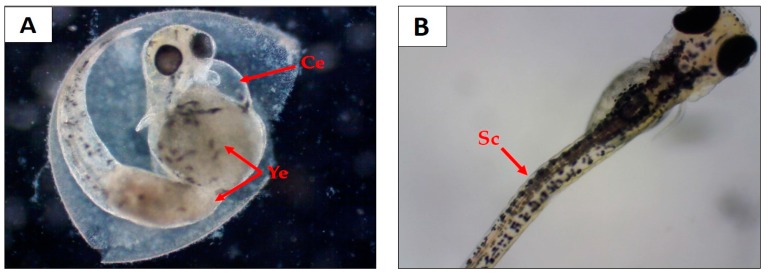
Teratogenic effects on *D. rerio* 96 hpf embryo. (**A**) Embryo treated at a concentration of 500 mg/L, where cardiac edema (Ce) and yolk edema (Ye) were observed; (**B**) embryo treated at a concentration of 250 mg/L, where scoliosis (Sc) was observed.

**Table 1 pharmaceuticals-12-00175-t001:** Identification of the constituents of the hydroethanolic extract of the fruits of *Libidibia ferrea* (EHEFLf) by LC-HRMS/MS.

RT (min)	[M-H]^−^	Molecular Formula	Error (ppm)	MS/MS	Substance
0.89	169.01340	C_7_H_6_O_5_	5.0	125.02324	Gallic acid
0.81	331.06729	C_13_H_16_O_10_	0.7	271.04614 241.03496 211.02391 169.01331 125.02332	Galloyl-glucose Ester
0.86	345.08279	C_14_H_18_O_10_	0.2	313.05731 169.01337 125.02334	Gallic acid methoxy glycoside
0.73	179.05540	C_6_H_12_O_6_	0.9	161.04422 131.03365 113.02316 95.01269	Hexose
1.15	483.07861	C20H20O14	1.2	331.0679 313.0551 271.0466 211.0238 169.0135	di-O-galloyl-d-hexose
1.94	633.07385	C27H24O18	0.8	463.0479 300.997	Corilagin
2.1	469.00504	C21H10O13	0.4	425.01523 299.99118	Unknown
3.98	197.04486	C9H10O5	3.5	169.01332 152.89384	Unknown
4.68	300.99905	C14H6O8	0.2	-	Ellagic Acid
6.25	449.10938	C21H22O11	1.1	287.055 151.002	Eriodictyol-O-hexoside
6.32	433.11426	C21H22O10	0.6	271.0593	Naringenin-O-hexoside

**Table 2 pharmaceuticals-12-00175-t002:** Distribution of the histopathological changes in *D. rerio* gills through the histopathological alteration index (HAI) described in levels.

Organ	Test Substance	Animal	Histological Changes		
			Level I	Level II	Level III
Gill	EHEFLf	01	7	0	0
		02	5	0	0
		03	6	0	0
		04	8	0	0
		05	6	0	0
Classification:	Normal			Mean: 6.4 ± 0.5	
			Level I	Level II	Level III
Gill	(Control-vehicle)	01	3	0	0
		02	3	0	0
		03	2	1	0
		04	3	0	0
		05	0	0	0
Classification:	Normal			Mean: 3.2 ± 0.3	

**Table 3 pharmaceuticals-12-00175-t003:** Distribution of the histopathological changes in *D. rerio* liver through the histopathological alteration index (HAI) described in levels.

Organ	Test Substance	Animal	Histological Changes		
			Level I	Level II	Level III
Liver	EHELF	01	5	3	0
		02	7	4	0
		03	7	5	0
		04	6	1	0
		05	10	3	0
Classification:	Moderate to severe			Mean: 39.0 ± 6.8	
			Level I	Level II	Level III
Liver	(Vehicle-control)	01	3	0	0
		02	8	1	0
		03	2	1	0
		04	2	0	0
		05	3	1	0
Classification:	Normal			Mean: 8.4 ± 2.4	

**Table 4 pharmaceuticals-12-00175-t004:** Distribution of the histopathological changes in *D. rerio* intestine through the histopathological alteration index (HAI) described in levels.

Organ	Test Substance	Animal	Histological Changes		
			Level I	Level II	Level III
Intestine	EHELF	01	8	0	0
		02	8	1	0
		03	9	1	0
		04	4	1	0
		05	7	1	0
Classification:	Mild to moderate			Mean: 15.2 ± 2.0	
			Level I	Level II	Level III
Intestine	(Vehicle-control)	01	1	1	0
		02	3	1	0
		03	3	1	0
		04	1	1	0
		05	3	1	0
Classification:	Normal			Mean: 12.2 ± 0.5	

**Table 5 pharmaceuticals-12-00175-t005:** Distribution of the histological changes in *D. rerio* kidneys by HAI according to the level of changes.

Organ	Test Substance	Animal	Histological Changes		
			Level I	Level II	Level III
Kidneys	EHELF	01	8	0	0
		02	8	1	0
		03	9	1	0
		04	4	1	0
		05	7	1	0
Classification:	Moderate to severe			Mean: 28.6 ± 2.6	
			Level I	Level II	Level III
Kidneys	(Vehicle-control)	01	2	0	0
		02	2	1	0
		03	3	0	0
		04	2	0	0
		05	2	1	0
Classification:	Normal			Mean: 6.4 ± 2.3	

**Table 6 pharmaceuticals-12-00175-t006:** Overview of teratogenic and morphological effects caused by EHEFLf concentrations (25, 50, 125, 250, and 500 mg/L) in the *D. rerio* embryos at 96 hpf.

Teratogenic Changes		mg/L						
	Propyleneglycol 1%	25	50	125	250	500	Σ_t_	%
Cardiac edema	0	0	0	0	0	20	20	11.1
Scoliosis	0	0	0	0	10	0	10	5.5
Yolk sac edema	0	0	0	0	5	21	26	14.4
Growth retardation	0	0	0	0	0	0	0	0
**Lethal embryos**	8	9	10	3	4	3	n/a	n/a
**Σ** Teratogenic embryos	0	0	0	0	15	21	36	n/a
**%** Teratogenic embryos	0	0	0	0	50	70	n/a	n/a
**%** Lethal embryos	26.6	30	33.3	10	13,3	10	n/a	n/a

n/a = not applicable.

**Table 7 pharmaceuticals-12-00175-t007:** Behavioral alterations of *Danio rerio*.

Stages	Behavioral Changes
I	Increased swimming activity; Spasms; Tail axis tremors
II	Circular swimming; Loss of posture.
III	Loss of motility; Deposition at the bottom of the aquarium; Death

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
