# Peer review of "Libidibia ferrea (jucá), a Traditional Anti-Inflammatory: A Study of Acute Toxicity in Adult and Embryos Zebrafish (Danio rerio)"

_pharmaceuticals, 2019, doi:10.3390/ph12040175_

Round 1

Reviewer 1 Report

   Using adult zebrafish and zebrafish embryos as toxicity test systems, the authors claimed that EHEELf reaches the acceptable safety level as an oral medicinal product. This submission can not be considered for publication unless a major revision has been accomplished. The points needed to be revised include (1) What components actually carry out the antiinflammatory effects of EHEELf? Table 2 lists only major components, but what substances generated signals at retention times 2.09, 3.98, and 4.66? (2) Adult zebrafish was orally administered with the extract powder at 2g/Kg and AHI showed the induction of pathological parameters in the gills, livers, and kidneys of these fish. If tissue damages are induced by the extract at this dose, how do we know that EHEELf is safe to human? An acute toxicity using rat or mouse as a model organism should be conducted to provide stronger evidences that EHEELf at doses used for human therapy are really safe. It is unclear why the authors present figure 10 showing edema caused by EHEELf. It is important to relate EHEELf at 250 mg/L to doses used for human treatment. The authors should be clear in mind that the objective of this paper is to demonstrate the safe use of EHEELf or to provide a warning not to overuse it. (4) How the oral administration of EHEELf to adult zebrafish should be described in detail in materials and methods. (5) An extensive revision of English language should be taken to make this article easier to read and to avoid misunderstanding.

Author Response

Dear Colleague,

The answer are attached. 

Reviewer 2 Report

The manuscript deals with the exposure related toxicity of Libidibia ferret (Juca) plant in Danio region embryos.Animals were exposed to hydroethanolic extract and changes in the organic-cellular level of the gills, liver, kidneys, and intestine and on embryos were investigated. The extract was also analysed by LC-M/MS, and the results suggest a high concentration of polyhydroxylated substances. The paper is well prepared and there is merit in the data; however it requires some corrections and clarifications prior to acceptance for publication.  

The scope of the study is not clear - the effects of hydroethanolic extract of fruits of Libidibia ferrea (EHEFLf) on zebrafish is investigated from an ecological perspective? Is it to address worries about environmental contamination? and the potential effects on wildlife? Or is this just a first step of investigative studies, using zebrafish as model organisms, to infer effects on humans? The scope has to be clear from the very start of the manuscript. 

The conclusion of this study is a bit deceiving - it sounds very clear cut: if there is no evidence of effects on zebrafish embryo (following a very short exposure, in a laboratory controlled experiment), then the authors concluded that the oral use of the fruit product is safe for humans! This should definitely be toned down. 

Legend to Fig 1 should be more detailed. 

Actually the authors are requested to review all Figure legends. I.e. Figure 3 - this one should be self explanatory (especially when the reader is offered the results before the methods): "Histological alterations of gills in Danio rerio exposed to......at a concentration of ......for....(days/hours)..... no animals exposed? treatment replicates?...

Results page 3, line 81: "slight alterations"  - very vague, at least some details needed in a bracket.

Figure 2: A control group is needed here : animals kept in perfectly clean environment. 

Table 3: "Distribution of histopathological changes by AHI according to the level of changes" - what is AHI? full name please. And the legend should avoid the repetition of the word Change!

Same for table 4 and 5.

Figure 4,5 - see comments for Fig 3.

Page 7 line 166 - this should be declared in the introduction, to set up the scope of the study

Page 8 line 212 - the sentence is too long and confusing, please reformulate

Page 10 line 267 - "The toxicity test consisted of two steps: 2 g/kg oral limit test following the technique...". Where is the second step??

And why 2g/Kg? the authors should be very clear in explaining their choice. 

Author Response

(The authors gave the same response as above.)

Round 2

Reviewer 1 Report

Pharmaceuticals-619038 has been carefully revised and Tab.1 now shows a much clearer identification of active compounds present in EHEFLf. In addition, the discussion has also be improved. However, nuclear atrophy (NA), not AN, should be corrected in the legend of Fig.4. The overall quality of this submission now meets the level of acceptance. 

Reviewer 2 Report

The manuscript is much improved, I recommend it for publication in the present form, please check spelling errors.